# Amplification Refractory Mutation System (ARMS)-PCR for Waxy Sorghum Authentication with Single-Nucleotide Resolution

**DOI:** 10.3390/foods10092218

**Published:** 2021-09-18

**Authors:** Xiaoying Zhu, Minghua Wu, Ruijie Deng, Mohammad Rizwan Khan, Sha Deng, Xi Wang, Rosa Busquets, Wanyu Deng, Aimin Luo

**Affiliations:** 1Healthy Food Evaluation Research Center and Key Laboratory of Food Science and Technology of Ministry of Education of Sichuan Province, College of Biomass Science and Engineering, Sichuan University, Chengdu 610065, China; zhuxiaoying0627@163.com (X.Z.); wuminghua06@163.com (M.W.); ds585455@foxmail.com (S.D.); wanyudeng@163.com (W.D.); 2Department of Chemistry, College of Science, King Saud University, Riyadh 11451, Saudi Arabia; mrkhan@KSU.EDU.SA; 3Sichuan Langjiu Group Co., Ltd., Luzhou 646523, China; 4School of Life Sciences, Pharmacy and Chemistry, Kingston University London, Penrhyn Road, Kingston upon Thames KT1 2EE, UK; r.busquets@kingston.ac.uk

**Keywords:** ARMS-PCR, DNA barcode, food authenticity, sorghum, liquor

## Abstract

Waxy sorghum has greater economic value than wild sorghum in relation to their use in food processing and the brewing industry. Thus, the authentication of the waxy sorghum species is an important issue. Herein, a rapid and sensitive Authentication Amplification Refractory Mutation System-PCR (aARMS-PCR) method was employed to identify sorghum species via its ability to resolve single-nucleotide in genes. As a proof of concept, we chose a species of waxy sorghum containing the wx^c^ mutation which is abundantly used in liquor brewing. The aARMS-PCR can distinguish non-wx^c^ sorghum from wx^c^ sorghum to guarantee identification of specific waxy sorghum species. It allowed to detect as low as 1% non-wx^c^ sorghum in sorghum mixtures, which ar one of the most sensitive tools for food authentication. Due to its ability for resolving genes with single-nucleotide resolution and high sensitivity, aARMS-PCR may have wider applicability in monitoring food adulteration, offering a rapid food authenticity verification in the control of adulteration.

## 1. Introduction

Sorghum is the fifth largest cereal crop in the world. It is the main food crop in Africa and the raw material for many famous liquor companies in Southwest China. Waxy sorghum is more suitable for liquor brewing, compared with wild sorghum. For instance, high-quality liquor enterprises such as Kweichow moutai, Lang Jiu and Wuliangye all rely on waxy sorghum. The choice of sorghum species is based on the fact that waxy sorghum improved the digestibility of starch and protein, in contrast to wild sorghum [1,2]. It was reported that waxy sorghum led to higher ethanol yield and higher conversion efficiency than wild sorghum [3,4]. And, its waxy starch had lower gelatinization temperature, which reduced the energy demand in the ethanol production process and shortened the fermentation time [5,6]. In addition, waxy sorghum is also widely used in the food processing industry [7,8]. Mezgebe et al. found that the waxy sorghum powder had considerable potential in the production of gluten-free fermented flat cake products, it had good structure and function [9]. The Elhassan group found that waxy sorghum flour, with high protein digestibility, had better dough-like properties than non-waxy sorghum, with normal protein digestibility [10,11].

The market price of wild sorghum is usually half that of waxy sorghum, and the waxy sorghum varieties may also lead to differences in flavor of the liquors and the quality of foods where they are used. As a consequence, these differences between various species open the door for food adulteration by substituting high-value species with low-priced ones, as large profit margins can be achieved [12,13]. A vast majority of waxy sorghum are wx^a^ or wx^c^ in Southwest China, and many famous liquor enterprises usually use wx^c^ as the main raw material for brewing high-quality liquor. So it is necessary to assess its authenticity to guarantee the integrity and quality of the species used [14].

Waxy sorghum is the recessive mutant of wild sorghum. At present, four kinds of waxy sorghum have been found, named as wx^a^, wx^b^, wx^c^, and wx^d^ [15,16]. Studies by Sattler showed that the wx^a^ allele had a large insertion (>4 kb) in the third exon of the SbGBSS. The wx^b^ allele had a missense mutation (+1G-to-T) in the eighth exon that changed glutamine 268 to a histidine. Yuangen Lu et al. identified another two waxy alleles, wx^c^ and wx^d^. The wx^c^ allele had a G deletion at the 5′ splicing site of the ninth intron, and the wx^d^ allele had a G to C mutation at the 3′ splicing site of the tenth intron. These single-nucleotide variants can be used as molecular markers to identify different sorghum species genetically.

In general, through physical-chemical methods, only wild sorghum and waxy sorghum can be distinguished separately. Therefore, the identification of waxy sorghum with different genotypes needs to be explored at the molecular level. At present, there are a few studies on the identification of different waxy sorghum. The current methods of choice for gene identification of waxy sorghum are Sanger sequencing and PCR [17,18]. The Sanger sequencing technique is considered to be very reliable and accurate, but has some drawbacks. For example, samples need to be sent to a certified testing agency, and it is time-consuming, usually taking 24 h to several days to finish the test. Some work used PCR to identify different waxy sorghum [19,20]. Usually, they cannot resolve a single nucleotide in gene sequences which often occur in different species of sorghum. Therefore, alternative methods are needed which allow an inexpensive and fast screening of relevant species with single-nucleotide resolution.

Amplification Refractory Mutation System (ARMS), known as allele-specific amplification, combined with PCR (ARMS-PCR), allows to identify gene barcodes with single-nucleotide resolution, and is widely used in clinic diagnosis for cancers [21]. However, its potential for food authentication has not been explored so far in a rapidly evolving food control scenario.

Herein, we introduce ARMS-PCR to identify sorghum species, and create a tool, termed authentication ARMS-PCR (aARMS-PCR), to identify sorghum species with single-nucleotide resolution. We chose a species of waxy sorghum containing wx^c^ mutation which is abundantly used in liquor brewing. The aARMS-PCR can distinguish non-wx^c^ sorghum from wx^c^ sorghum to guarantee the identification of specific waxy sorghum species. It allowed to detect as low as 1% non-wx^c^ sorghum in sorghum mixtures, which are one of the most sensitive tools for food authentication. The aARMS-PCR would find wide applicability in food adulteration considering its ability to resolve genes with single-nucleotide resolution and high sensitivity [22,23].

## 2. Materials and Methods

### 2.1. Materials and Reagents

‘Hongyingzi’ was provided from the Sichuan Langjiu Group Co., Ltd. (Luzhou, China), and ‘Jinnuoliang No.6’ and ‘Jin 204’ were obtained from Sichuan Academy of Agricultural Sciences (Chengdu, China). Ezup Food Genomic DNA Extraction Kit (cat. No. GA15KA7284), SanTaq Plus PCR Master Mix (cat. No. B532071) and 50×TAE buffer (cat. No. B548101) were all bought from Sangon (Shanghai, China). The 2×PerfectStartTM Green qPCR SuperMix (cat. No. B639275) and nuclease-free H_2_O were bought from Tiangen (Beijing, China). DNA primers were synthesized by Sangon (Shanghai, China), and were purified using Polyacrylamide Gel Electrophoresis (PAGE).

### 2.2. Iodine Test

Wild sorghum contains 20–30% amylose, while waxy sorghum mainly contains amylopectin. Amylopectin is reddish-brown after reacting with iodine, in contract, amylose becomes dark blue. They react with iodine presenting different colors because of differences in their degree of polymerization or the relative molecular weight of starch. When the degree of polymerization of amylose is 200–980 or the relative molecular weight is 32,000–160,000, the color of the inclusion compound produced by the reaction of starch and iodine is blue. The average degree of polymerization of amylopectin with many branches is 20~28, and the inclusion compound is purple or even reddish-brown. Therefore, according to the principle, waxy and wild sorghum can be preliminarily identified by iodine staining method. The waxy seeds are reddish-brown after reacting with iodine solution, and the wild seeds are dark blue. Each cultivar was carried out three times.

### 2.3. DNA Isolation and Sequencing

Two pairs of primers [16] were used to sequence sorghum seeds to determine their genotypes. The first pair primers were Seq-F1 (ACCCAAACCCAGTACAAGGATAAG) and Seq-R1 (ACGGGCTTCTCGTAGCTGCAATC). The primers were to amplify the 5′ region of SbGBSS gene in sorghum. The second pair primers were Seq-F2 (TGGCATC TACAAGGACGCAAAG) and Seq-R2 (TTCTACACGCGATATCAGGGGTC) Which were to amplify the 3′ region of SbGBSS of genomic DNA.

All DNA extraction procedures were conducted using an Ezup Food Genomic DNA Extraction Kit according to the manufacturer’s instructions. Briefly, 100–200 mg sorghum powder, 1 mL GMO buffer and 20 μL proteinase K were added into the centrifuge tube. These were incubated at 65 °C for 30 min and centrifuged at 12,000 rpm for 10 min. After that, 400 μL chloroform was added to the supernatant, and the mixture was centrifuged at 12,000 rpm for 10 min. The new supernatant was transferred to a new centrifuge tube. The experimenter added GMP buffer into the new centrifuge tube, centrifuged it at 12,000 rpm for 5 min, then discarded the supernatant and retained the sediment. Then, 350 μL DRL buffer and 300 μL anhydrous ethanol were added to the sediment. All of them were transferred to the adsorption column after the sediment was fully suspended. Next, the adsorption column was centrifuged at 10,000 rpm for 2 min. Following, the solution was poured out in a collection tube. 500 μL wash solution was added to the adsorption column, and it was centrifuged at 10,000 rpm for 1 min. The experimenter needed to repeat this operation twice. Then, the solution in the collection tube was poured out again, and the adsorption column was centrifuged at 12,000 rpm for 2 min. Finally, the experimenter took out the adsorption column, put it into a new centrifuge tube, added 30 μL of nuclease-free H_2_O or TE buffer, and centrifuged it at 12,000 rpm for 2 min. The quantity and quality of the obtained DNA solution were measured with a microplate reader Synergy H1 (BioTek, Winooski, VT, USA) and stored at −20 °C for subsequent experiments.

### 2.4. Primer Design and PCR Amplification

Ten allele-specific forward primers and reverse primers were designed to distinguish between wx^c^ and non-wx^c^. The primers are listed in Appendix A.

The PCR reactions were performed in a final volume of 20 μL and contained 1 μL of DNA template, 1 μL of forward primer (0.2 μM), 1 μL of reverse primer (0.2 μM), 10 μL of SanTaq Plus PCR Master Mix, and 8 μL of nuclease-free H_2_O. The PCR conditions were proceeded as follows: an initial denaturation at 95 °C for 10 min; followed by 35 cycles of 95 °C for 10 s, 60 °C for 1 min, 72 °C for 30 s; and a final extension at 72 °C for 7 min. PCR amplifications were performed in a ProFlexTM 3 × 32-well PCR System (Thermo Fisher Scientific, Waltham, MA, USA).

The 5 μL of PCR products in 1 μL of 1 × gel loading buffer were used for gel electrophoresis. The gel electrophoresis was performed on the prepared gel (3% agarose-TAE sol-gel with 1 × GelRed dye) in 1 × TAE at 150 V for 30 min. After electrophoresis, the gel was visualized via Gel Doc XR+ system (Bio-Rad, Hercules, CA, USA).

### 2.5. aARMS-PCR Detection

A qPCR was performed using a qTOWER3G in a 96-well plate (Analytik Jena AG, Shanghai, China). The allele-specific, qPCR reactions were conducted in a final volume of 20 μL and that contained 1 μL of DNA template, 1 μL of forward primer (0.2 μM), 1 μL of reverse primer (0.2 μM), 10 μL of 2 × PerfectStartTM Green qPCR SuperMix, and 8 μL of nuclease-free H_2_O. The qPCR procedure involved initial activation steps of 50 °C for 2 min and 95 °C for 10 min, then followed by 40 cycles of 95 °C for 10 s, 60 °C for 1 min and 72 °C for 30 s.

### 2.6. Standard Curves

The DNA extracted in step 2.3 was diluted by nuclease-free H_2_O to a concentration of 100 ng/μL. It was then diluted in a continuous tenfold gradient to a minimum concentration of 1.0 pg/μL. The standard curve of wx^c^ was generated by plotting the cycle threshold (C_T_) values of the qPCR amplifications from different amounts of DNA (100 ng, 10 ng, 1 ng, 0.1 ng, 0.01 ng and 0.001 ng). The C_T_ value of each dilution corresponded to the log of the DNA concentration. Meanwhile, the gradient diluted wx^c^ DNA templates were subjected to PCR reaction for electrophoresis analysis.

### 2.7. Quantitative Detection in Mixed Cereal Grain Samples

Mixed DNA samples were prepared using various proportions of DNA templates extracted from sorghum (0, 1, 5, 10, 20, 50, and 100% non-wx^c^ DNA, with the remainder from wx^c^ DNA). The qPCR amplification of DNA mixtures was carried out in triplicate.

## 3. Results and Discussion

The prominent feature of aARMS-PCR lies in its ability to quantitatively detect known mutations with high specificity. The principle of aARMS-PCR for discriminating sorghum species via single-nucleotide identification is illustrated in Figure 1. The wx^c^ and non-wx^c^ allele-specific forward primers and a common reverse primer were designed according to wx^c^ SNP site. When the 3′ end base A of the wx^c^ allele-specific primer matched the wx^c^ SNP base T, the primer started to extend from the 3′ end of the primer to generate the wx^c^ allele-specific products. In contrast, the 3′ end base A of the wx^c^ allele-specific primer did not match the non-wx^c^ SNP base G, the primer extension was terminated. In general, it is not ideal to distinguish alleles by a single base mutation at the 3′ end of the primer. Thus, a mismatched base is added to the second or third at the 3′ end of the primer for enhancing the specificity of primers [5,24]. This additional mismatch destroys the stability of the pairing between the primer and the non-corresponding template and reduces the amplification efficiency of the non-corresponding template [25]. Thus, for the non-wx^c^, when the mismatched base was placed fourth from the 3′ end of wxt F4 (substitution of C for T), aARMS-PCR yielded the highest discrimination ability between wx^c^ and non-wx^c^ genes. For wx^c^, while the mismatch (substitution of A for C) was introduced second from the 3′ end of wxc F4, the wxc F4 could specifically identify wx^c^ waxy sorghum (Appendix A).

### 3.1. Method Validation

In order to investigate the ability of aARMS-PCR for the discrimination between wx^c^ waxy sorghum and non-wx^c^ sorghum, three sorghum cultivars were firstly sequenced. The sequencing results are shown in Figure 1A and Appendix A. The results indicated that ‘Hongyingzi’ was wx^c^, and ‘Jinnuoliang No.6′ was wx^b^, and the ‘Jin 204’ was wild.

Then, allele-specific primers were used for qPCR amplification of ‘Hongyingzi’ and ‘Jinnuoliang No.6’ to assure the authenticity of ‘Hongyingzi’ in the waxy sorghum mixture. A total of 400 ng of the DNA extracted from sorghum was used for aARMS-PCR tests. The qPCR amplification results are shown in Figure 1C,D. C_T_ values of ‘Hongyingzi’ amplified by wxc F4 and wxt F4 were 22.59 and 27.58, respectively. The C_T_ of ‘Jinnuoliang No.6′ amplified by wxt F4 and wxc F4 were 19.95 and 31.95, respectively. The results indicated that wxc F4 and wxt F4 allowed to differentiate the ‘Hongyingzi’ from the ‘Jinnuoliang No.6′ species.

To verify the accuracy of aARMS-PCR, we performed sorghum iodine staining and PCR experiments. As shown in Figure 1B, ‘Hongyingzi’ and ‘Jinnuoliang No.6’ were reddish brown after iodine staining, while ‘Jin 204′ was dark blue. Therefore, ‘Hongyingzi’ and ‘Jinnuoliang No.6′ were waxy sorghum, and ‘*Jin 204*’ was wild sorghum. This was consistent with the aARMS-PCR results. The primers wxc F4 and wxt F4, with ‘Hongyingzi’ and ‘Jinnuoliang No.6’, were paired for PCR reaction. In order to highlight the specific amplification of ‘Hongyingzi’ and ‘Jinnuoliang No.6’ by specific primers, wxc F4 and wxt F4 respectively, the amount of DNA used in PCR was reduced to 50 ng, compared with the 400 ng of DNA used in qPCR. The electrophoretic results are shown in Figure 1E. According to the gel imaging results, the ‘Jinnuoliang No.6’ amplified using wxt F4 and ‘Hongyingzi’ amplified using wxc F4 yielded bright bands in the corresponding lanes. The location of the bands confirmed the expected sizes of the DNA fragment. In contrast, the ‘Jinnuoliang No.6′ amplified by wxc F4 and ‘Hongyingzi’ amplified by wxt F4 were dark in the corresponding lanes. This may be due to the dual effects of the decreased amount of DNA and the lower sensitivity of PCR and electrophoresis than qPCR. Therefore, wxt F4 identified the ‘Jinnuoliang No.6’, and wxc F4 identified the ‘Hongyingzi’. This was consistent with the aARMS-PCR results.

### 3.2. Sensitivity of aARMS-PCR

To determine the sensitivity of aARMS-PCR using PCR and qPCR based on the allele-specific wxc F4, DNA samples were obtained from ‘Hongyingzi’ grains (100 ng, 10 ng, 1 ng, 0.1 ng, 0.01 ng and 0.001 ng). The samples for qPCR were used to construct the standard curves for quantification of target genes. The detection limit of PCR was down to 1 ng according to the electrophoresis results (Figure 2A). No obvious band was found with an input of DNA templates lower than 1 ng. The curve was linear over a range of 100.0 ng to 1.0 pg, and the detection limit of wxc F4 was approximately 1.0 pg (Figure 2B and Appendix A). The sensitivity of aARMS-PCR was dramatically enhanced via the replacement of PCR-gel imaging by qPCR.

### 3.3. Quantitation of Adulteration of Sorghum Species Using aARMS-PCR

The capacity of aARMS-PCR to quantify sorghum adulterations was evaluated using sorghum mixtures. DNA mixtures that contained different proportions of ‘Jinnuoliang No.6’ (0%, 1%, 5%, 10%, 20%, 50% and 100%) in mixed ‘Jinnuoliang No.6’ and ‘Hongyingzi’ were prepared. An amount of 300 ng of each DNA mixture was then subjected to two separate aARMS-PCR reactions using wxt F4 and wxc F4 along with the R1. Its amplification curve was shown in Appendix A. ∆C_T_ was defined as the difference between C_Twxt F4_ and C_Twxc F4_, to estimate the ratio of products produced by reaction with wxt F4 and wxc F4. The quantification performance using aARMS-PCR was validated by using a series of different concentrations of ‘Jinnuoliang No.6’ in the mixture. Further, to investigate the detection of aARMS-PCR at low concentrations of ‘Jinnuoliang No.6’ in mixed grains, a histogram of ‘Jinnuoliang No.6’ at low concentrations (0%, 1%, 5% and 10%) was established. The linear regression equation of the ∆C_T_ was calculated as A = −0.144 × B + 3.4855 (R^2^ = 0.9971) (linear dynamic range, 0–100%), where A and B represented the value of the ∆C_T_ and the content ratio of the ‘Jinnuoliang No.6’ in the mixture, respectively. ∆C_T_ (C_Twxt F4_-C_Twxc F4_) were greater than 2 at low concentrations (such 1% and 10%), suggesting the aARMS-PCR can endow the ability of quantifying ‘Jinnuoliang No.6’ in mixed grains. Significant differences as indicated by ∆C_T_ between ‘Jinnuoliang No.6’ with 0% and 1% content were observed (*p* = 0.002 < 0.01), thus reaching a very significant level. We also formed a standard curve and a histogram of ‘Jinnuoliang No.6’ with different content ratios amplified by wxt F4 (Figure 3B,D). By comparing ∆C_T_ and C_Twxt F4_, the data showed that the ∆C_T_ had better resolution at low concentration of ‘Jinnuoliang No.6’ in the mixture. This developed method can detect satisfactorily the authenticity of wx^c^ and help to prevent adulteration of non-wx^c^ sorghum.

## 4. Conclusions

In summary, we introduced an aARMS-PCR method for the discrimination of adulteration in sorghum samples. The assay presents novel features: (1) specific recognition, the assay can identify species with single base differences; (2) high sensitivity: aARMS-PCR can detect 1.0 pg wx^c^ waxy sorghum DNA, compared to the common PCR, which could only detect 1.0 ng DNA. In addition, aARMS-PCR can detect as low as 1% non-wx^c^ sorghum in mixtures including wx^c^ waxy sorghum; (3) This is a potential universal platform: the successful applications of this assay for determining wx^c^ waxy sorghum and non-wx^c^ sorghum indicate its robustness for quantifying samples. Therefore, aARMS-PCR can serve in routine food integrity and authenticity tests.

## Data Availability

Not applicable.

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
