# Peer review of "Amplification Refractory Mutation System (ARMS)-PCR for Waxy Sorghum Authentication with Single-Nucleotide Resolution"

_foods, 2021, doi:10.3390/foods10092218_

Round 1

Reviewer 1 Report

This paper indicates  ARMS-PCR to identify sorghum species with single-nucleotide resolution. It is not a special technique for detecting single nucleotide polymorphisms, but it is  important technique for food quality control. 

The design of the study is very good, but it seems that the Ct values of qPCR and the results of electrophoresis are inconsistent. 

In Fig.1(C), the result for N4 of qPCR seems to be almost saturated as well as that of N1 and N3. In that case, the amount of amplification product depends on the amount of primer, so it is considered that a considerable amount is formed. Why is the band not visible at all in the result of electrophoresis in Fig.1 (E)?

Reviewer 2 Report

- The article is not in the Food journal format. Lines are not numbered off, the page number is not correct and the headlines have not been employed. Please, use the Food template that you can find in "Instructions for Authors" from the Food webpage. -The introduction provides enough information about the adulteration issue of waxy and wild sorghum. The authors introduce the topic and the use of these sorghums and the reason because wild sorghum is usually adulterated with waxy sorghum. The techniques usually employed for the detection of adulterations have been included in the introduction as well. However, important lacks in the English language have been detected. The authors employed informal expressions, long paragraphs, and even incorrect words/expressions. For example, line 7: contrasted; Line 10: meanwhile, or in the last paragraph "herein, introduce..." -The materials and method section provides enough information about the materials and reagents used. • In the iodine test extra information must be provided about the differences in the iodine results test for distinguished according to the waxy or wild variety and the reason for this difference. • Points 2.3, 2.4, and 2.5: information required is included but again, the process description is disorganized, informal expressions are used and it is very difficult to follow. • Point 2.6: extra information about the preparation and the results of standard curves employed must be included. • Point 2.7: how has been the DNA template extracted? and which procedure has been followed for the qPCR amplification?. Please include all this information. -In the case of the results and discussion section, I consider that the results are in accordance with the aims of this research. However, first of all, the English language continues to be very poor and difficult to follow, and after that, I can not see an advantage in using this methodology from other different techniques faster, and cheaper. Can the authors include an extra paragraph clarifying the advantages of this method?

Round 2

Reviewer 2 Report

I consider that the changes did for the authors respond to the lacks previously detected, enhancing the quality of the article and providing enough information to other researchers can reproduce the experiment.
I accepted in the present form.
